# Stakeholders' Social Network in the Participatory Process of Formulation of Natura 2000 Management Programme in Slovenia

**Tomislav Laktić [1], Aleš Žiberna [2], Tina Kogovšek [2] and Špela Pezdevšek Malovrh [3],***

[1]  Water and Investments Directorate, Cohesion Division, Ministry of the Environment and Spatial Planning, Dunajska cesta 48, 1000 Ljubljana, Slovenia; tomislav.laktic@gov.si

[2]  Faculty of Social Science, University of Ljubljana, Kardeljeva ploščad 5, 1000 Ljubljana, Slovenia; ales.ziberna@fdv.uni-lj.si (A.Z.); tina.kogovsek@fdv.uni-lj.si (T.K.)

[3]  Department of Forestry and Renewable Forest Resources, Biotechnical Faculty, University of Ljubljana, Večna pot 83, 1000 Ljubljana, Slovenia

\*  Correspondence: spela.pezdevsek.malovrh@bf.uni-lj.si

**Abstract:** Stakeholder participation has become an important driving force in policy decision-making and implementation, particularly in the nature conservation sector, where complex interactions and conflict of interest between stakeholders are common. A stakeholder analysis, which was complemented with a social network analysis, was used to examine the cooperation and conflict network between stakeholders, their institutions, and sectors in the case of the formulation of the Natura 2000 Management programme in Slovenia for the period 2015–2020 (PUN). Using data from a web survey (*n* = 167), cooperation and conflict networks were analysed while using degree centrality, indegree centrality, betweenness centrality, and blockmodeling. The results of the stakeholder analysis showed that the highest number of stakeholders that are involved in the participatory process of PUN was from the forestry and hunting sector, followed by the agriculture and nature conservation sector. The results of the cooperation network showed that the network is highly centralized, with only few institutions taking a central position in the PUN process (Institute for Nature Conservation, Ministry of Environment and Spatial Planning, Chamber of Agriculture and Forestry, Ministry of Agriculture, Forestry and Food, and the Slovenian Forest Service). Moreover, the nature conservation sector was, on average, a sector with the highest concentration of power. In addition, in the cooperation network, which was fragmented across sectors, there were institutions that belonged to the same sector, which tended to cooperate with each other. The analysis of the conflict network showed that institutions with a central position in the cooperation network also had a central role in the conflict network. In addition, conflicts between institutions more frequently appeared among institutions from different sectors. The exceptions were institutions from the fishery and water sector, as this sector seemed to have many conflicts within it. Based on a blockmodeling, four groups of institutions were identified according to their cooperation network (core institutions, semi-core institutions, semi-periphery institutions, and periphery institutions). Our finding suggested that the participatory process of formulating PUN needs to be improved in such a way that in the future various stakeholders, especially excluded local ones, are more actively involved and a balance of the power between the stakeholders involved achieved.

**Keywords:** social network analysis (SNA); cooperation and conflict networks; stakeholders' involvement; participatory process; Natura 2000 management Programme

## 1. Introduction

Natura 2000 is the core pillar of European Union's (EU) biodiversity conservation policy [1–3]. It refers to an EU-wide ecological network of protected areas that extends across national borders, administrative levels, policy sectors, and socio-economic contexts [4]. The network is established and managed according to the legally binding provisions of the EU's Birds and Habitat Directive [5,6]. The Directives are transposed into national legislation, but the EU gives member states the freedom to choose the most appropriate means to achieve their goals. The Directives do not require management plans to be drawn up for Natura 2000 sites, but the Habitat Directive recommends their use as a means of ensuring the conservation status of sites [7,8]. The preparations of sites-level management plans are promoted in most member states as the main tool to identify conservation measures at site level despite soft regulation [9]. Additionally, the Directives do not require public or stakeholder participation in the process of management planning, although the guidelines emphasize the importance and benefits of such participation [10]. In other words, the Directives indirectly recognize the importance of public participation and the need for stakeholder involvement in the establishment and management of Natura 2000 sites [8,11–14].

The implementation of the objectives of the directives at the national level has changed the power and relationships between stakeholders in the decision-making process with the involvement of new stakeholders in the nature conservation system [7,15,16]. National legislation defines the role of the main stakeholders in nature conservation and establishes a multi-level governance system of Natura 2000 [16–18]. In addition, in the implementation of the nature conservation policy, the inter-institutional cooperation between different stakeholders from different sectors (e.g., forestry, agriculture, fishery, and nature protection) is a key factor for the success of participatory decision-making process across several jurisdictional levels [19–21].

The process of designation, implementation, and management of the Natura 2000 sites is complex and cumbersome process. A large number of stakeholders and institutions have a direct or indirect role in these processes regarding the land designated as Natura 2000 sites [22–27]. However, there are no regulatory rules on stakeholder participation [28,29], as the Habitats Directive does not clearly establish participatory approaches. It states that conservation measures shall take into account human (economic, social and cultural) needs and local characteristics, but site designation is only based on (ecological) scientific criteria, while social criteria are not even mentioned. For this reason, participatory approaches were initially ignored in many EU member states and technocratic approaches dominated, privileging conservation experts and marginalizing socio-economic stakeholders [10,13,14,24]. In addition, the numerous conflicts that were related to the Natura 2000 sites designation, implementation, and management emerged as a result of the absence of stakeholders' involvement and participation in these processes [13,19,30–33]. The identified conflicts were related to the conflicting stakeholders' interests, values, and perceptions, as well as to different and competing land use principles [14,19,26,31,34–40]. The involvement of and cooperation with various stakeholders as well as stakeholder participation and coordination between institutions was highlighted as an important instrument for increasing the acceptance of Natura 2000 in order to prevent further conflicts and improve the implementation and thus the conservation results [41].

According to Elsasser [42], several degrees of participation are possible, ranging from »passive« participation (with the modest claim that stakeholders are informed about the decisions made by others, and the decision-making process thus become transparent) to »interactive« participation, which requires a joint decision and perhaps shared liability. Passive and interactive participation can be seen as two poles of a continuum. In most cases, the process of designation, implementation, and management of the Natura 2000 sites will tend to follow an approach with different possibilities for stakeholders to influence the outcome.

For this reason, stakeholder analysis has gained increasing attention in nature conservation policy and it is now an integral part of participatory processes [43]. Many authors have highlighted he importance of stakeholder analysis [43–48], as it is a technique that intends to identify all groups of

stakeholders, organised or not, who have a common interest in a particular issue, the conflicts of interest between them, and the possible coalitions [49]. In addition, stakeholder analysis enables us to identify the key stakeholders and reveal their role, intentions, connections, interests, behaviour, influence, power, and position that they have in the decision-making process [46,50]. Stakeholder analysis in the case of Natura 2000 mainly focuses on participation in the implementation of Natura 2000 [14,23,31,51,52], while few studies focus on stakeholder analysis in the case of the formulation of Natura 2000 management plans [9,14,21].

Social network analysis (SNA) has often been used to identify interactions between stakeholders or institutions based on the role and influence they have in their networks, as it usually deals with the connectivity and interactions between stakeholders or institutions, to enhance stakeholder analysis [53]. Importantly, the SNA approach can reveal the position of each stakeholder participating in the network and it can also help in optimizing the flow of information [54]. The SNA has been widely used in nature conservation-related studies [22,29,47,55–58], yet there has been limited focus on the specific interactions among stakeholders (cooperation and conflicts) in the case of the formulation of the Natura 2000 management programme. However, the studies focusing on cooperation, see e.g., [11,37,56], reported that the participatory processes were less participatory than expected, being centralized around a small number of public authorities, with the low involvement of NGOs and private stakeholders. The pre-existing power of public authorities probably inhibits the ability of NGOs to collaborate with private stakeholders. Moreover, studies also revealed a lower level of cooperation of stakeholders in the network with other institutions, which indicates a clear top-down approach to the participatory process. To the best of our knowledge, the existing studies on the formulation of the Natura 2000 management programme fails to study conflict networks between stakeholders.

In order to fill the above mentioned gaps, this paper aims to analyse the participatory process of the formulation of the Natura 2000 Management Programme for the period 2015–2020 in Slovenia (PUN) using stakeholder analysis in combination with the SNA in order to (a) identify the main institutions in cooperation and conflict network and analyse their position and power in this network structure; (b) analyse sectors' involvement in the formulation of PUN and frequency of their cooperation and conflicts; and, (c) to cluster institutions based on cooperation network using a blockmodeling approach.

The results could be useful in informing the institutions that are responsible for Natura 2000 planning and management, as well as policy decision-makers about the failure of the existing participatory process and thus improve the quality of future processes. Furthermore, the results may be useful for policy decision makers at the national and EU level to develop guidelines for such participatory processes.

## 2. Materials and Methods

*2.1. Stakeholders' Involvement in the Process of the Formulation of Natura 2000 Management Programme (2015-2020) in Slovenia*

The process of the formulation of PUN started in 2012. The main focus was on: the preparation of detailed conservation objectives for Natura 2000 sites in Slovenia; the identification of measures to achieve conservation objectives, which are implemented in sectoral management plans (forestry, hunting, fishery, and water sector); and, those responsible for their implementation—institutions, which are responsible for the planning and implementation of nature protection measures in accordance with Slovenian legislation. The activities necessary for the adoption of this operational programme were supported by the LIFE + project 11 NAT/SI/880, whose coordinating partner was the Ministry of the Environment and Spatial Planning (MESP—responsible for the preparation of legislation regarding the environment) and whose project partners were the Institute of the Republic of Slovenia for Nature Conservation (IRSNC—responsible for nature conservation), the Slovenia Forest Service (SFS—responsible for forest management planning for state and private forests and elaboration of regional hunting management plans), the Fisheries Research Institute of Slovenia (FRIS—responsible for preparing fishery management plans in fishing areas), the Institute for Water of the Republic of Slovenia

(IWRS—was responsible for water related land management), and the Chamber of Agriculture and Forestry of Slovenia (CAFS—the organisation which represents all natural persons/private individuals and legal entities from the fields of agriculture, forestry, and fishing in the Republic of Slovenia; their employees provide services in agriculture and forestry extension). The Slovenian Water Agency from 2015 combines the implementation of professional, administrative, and developmental tasks, which were carried out by the Institute for Water of the Republic of Slovenia during the time of the participatory process of designation PUN.

The communication plan was prepared at the beginning of the process by an outsourced company and the project partners. The plan included a stakeholder analysis, different ways of involving stakeholders from different sectors at each stage with the aim of providing information, consultation, and participatory decision-making on the management of Natura 2000 sites in the future. The first draft of PUN 2015–2020 was prepared in collaboration with the project partners and additional experts on habitat types (forest or grassland habitats, etc.), and plant and animal species (birds, bugs, amphibians, etc.) from different institutions (e.g., universities, institutes, and NGOs). The draft was discussed with key stakeholders during six targeted roundtables (public meetings) in different parts of the country with representatives of different sectors (forestry and hunting, agricultural, fishery and water, nature conservation, and "others") (see Table 1). The amended draft of PUN was the subject of intragovernmental consultation, which consisted of consultation meetings with all ministries and their public bodies, as well as the Chamber of Commerce. The PUN 2015–2020 draft was adopted on April 2015. Following its adoption, eight workshops were organized for stakeholders, representatives of the above-mentioned sectors, and management organizations of nature parks to spread the information about PUN.

**Table 1.** Division of stakeholders by sector.

| Sector | Number of Participants in PUN | % | Number of Respondents | % |
|---|---|---|---|---|
| Forestry and hunting | 181 | 22.20 | 68 | 40.72 |
| Agriculture | 366 | 44.90 | 45 | 26.95 |
| Fishery and water | 34 | 4.13 | 8 | 4.79 |
| Nature conservation | 169 | 20.77 | 39 | 23.35 |
| "Others" | 65 | 8.00 | 7 | 4.19 |
| Total | 815 | 100.00 | 167 | 100.00 |

PUN: Natura 2000 Management programme in Slovenia for the period 2015–2020.

In the elaboration of the PUN, there has been a switch from increased information-communication with stakeholders in the past (gathering objective information to understand the problem, alternatives opportunities and/or solution) to consultation with key stakeholders (stakeholder feedback on analysis, alternatives, and/or decisions) although the adopted approach was still a top-down approach [19].

*2.2. Data Collection*

The study is based on a web questionnaire with a link being sent via e-mail using the 1KA web survey program (https://www.1ka.si) to all identified stakeholders who have participated in PUN. A preliminary list of stakeholders involved in PUN was drawn up on the basis of a list of participants in the workshops and reports of LIFE + project 11 NAT/SI/880. Eight hundred and fifteen stakeholders were identified as a study population. For 48 stakeholders, the contacts could not be found and were therefore excluded from the study. Therefore, the questionnaire was sent to 767 identified stakeholders by e-mail. The stakeholders came from different institutions (i.e., ministry, public forest administration, NGOs, university and research institutions, private forest owners, and farms associations) and they were divided into five main groups according to the sector to which they belong to (forestry, agriculture,

water and fishery, nature protection, and "other"—including the spatial planning and energy sector, regional development agencies) (Table 1). Dillman's Tailored Design Method [59] was adopted in order to maximize response rates and reduce survey errors. Two reminders were sent to those who had not replied within two and four weeks after the original deadline. The response rate was 34.8%, with 266 completed questionnaires, 99 of which were not suitable for the SNA, because the respondents skipped answering this part of the questionnaire (incomplete survey). Consequently, 167 questionnaires were used for the SNA.

The questionnaire consisted of six sections seeking information on: (1) nature protection policies; (2) the participatory process and the relations between stakeholders; (3) the influence of stakeholders on the process; (4) SNA of stakeholders in a cooperation network; (5) conflicts and SNA of stakeholders in a conflict network; and, (6) the socio-demographic characteristics of stakeholders. Two sections focused on stakeholders' networks in a PUN process that consists of two parts—cooperation network and conflict network. In the fourth section, a question was asked with that aimed to gather information regarding the stakeholders' cooperation in a network. In the question, the respondents were asked to identify the institutions they contacted during the participatory process from a drop-down list of institutions. The data that were gathered for creating and analysing the cooperation network were measured at the individual level of a stakeholder and then aggregated at the level of organizations/institutions. The data for creating and analysing the conflict network included a question from section five. The respondents were asked whether they had noticed a conflict and if, from the drop-down list of institution, they selected which institutions were involved in the conflict.

## 2.3. Data Analysis

The data that were obtained from the survey related to stakeholders' cooperation and conflicts in the formulation of PUN were transferred into a matrix scheme and used for the SNA. The conflict network is based on the question, whether the person has noticed a conflict and, if so, which institutions were involved in the conflict. The person could identify several conflicts. No meaningful direction can be deduced since there is no indication of the direction here and the person's institution could not be involved in the conflict, and therefore the conflict network is treated as an undirected network. On the other hand, the cooperation network is based on the question, with which institutions a certain person has cooperated (these answers were then attributed to the institution of this person). In the SNA it is customary to direct the tie in such cases (when these are perception based ties) from the person reporting cooperation to the one being reported to be involved in one, although cooperation is, in its core, undirected. Such coding reduces the bias due to the different types of reporting. An undirected network was only chosen for cooperation for the graphical representation (disregarding the direction of ties—who started communication/contact) to simplify the illustration, since only such a representation enables only drawing one tie between each connected institutions. In all other analysis, the cooperation was treated as a directed network. The centrality of the institutions and the blockmodeling solution were analysed in order to describe the general aspects of the cooperation and conflict network.

The position of an individual institution in the network was analysed while using two measures of centrality (degree and betweenness). The structural importance of an institution is usually assessed by the degree of centrality (DC), which takes the ties that an institution shares directly with another institution into account [60]. In other words, DC is defined as the number of institutions that are in direct contact with a particular institution and that have the capacity to directly communicate with others [61]. In a directed network, a DC distinction into indegree centrality (IDC) can be made. In present study, the IDC was only calculated for the cooperation network and not for the conflict related network, since only the cooperation network was directed. In our case, the IDC is related to the concept of prestige and it depends on the number of incoming ties. The IDC is the number of ties that an institution has that have been initiated by other institutions and can be used to estimate the importance of a particular institution in the social network. An institution is considered to be

prestigious if it is particularly visible to the stakeholders in the network; this means that the others recognize the institution.

The Betweenness centrality (BC) is calculated as the proportion of the shortest paths between the node pairs that pass through the node of interest [62] and measures the influence that an institution has on the dissemination of information in the network. Therefore, it identifies those institutions that play the role of intermediaries in the decision-making process [63]. Thus, these institutions have power in controlling information. According to Borgatti, et al. [64], the interpretation of BC works well for many social relations, but falls apart when we consider negative ties, such as conflict. It is difficult to know what to make of measures like BC when applying them to negative networks. In this study, BC has not been used for conflict network due to the above-mentioned facts.

The sum of squares homogeneity generalized blockmodeling with only complete blocks was used to find groups and ties between institutions in the directed cooperation network [65]. Blockmodeling is a technique that partitions the network in such a way that blocks, that is, ties between two clusters of institutions (or between institutions of the same cluster), follow a certain pattern. The network includes both institutions that participated in the study and those that did not. These later institutions cannot have outgoing ties in the network, as they did not have the opportunity to report them. They represent a kind of missing data. A blockmodeling approach for linked networks was used to blockmodel the entire network, keeping these two groups separate, that is, institutions, that participated in the survey, and those, that did not, could not be placed in the same cluster, in order to find groups and ties between institutions in the directed cooperation network. The number of groups was selected based on the review of solutions at different number of clusters and was set at 4 for participating institutions and 3 for non-participating institutions in a network.

The graphical elaboration and the main statistical features, network centralities, and blockmodeling were realized with the social network software Pajek [66–68] and the R statistical program [69], mainly using SNA packages [70], igraph [71], and blockmodeling [72].

## 3. Results

### 3.1. Basic Information about Institutions in a Network - Stakeholders' Analysis

The survey results of the PUN participatory process for the SNA included 167 stakeholders from different institutions and sectors, which were, for the purpose of our stakeholder analysis, categorized into five groups by sectors, according to the categorisation that was done in PUN: forestry and hunting sector (yellow vertices in Figure 1), the agriculture sector (green vertices in Figure 1), the fishery and water sector (blue vertices in Figure 1), the nature conservation (purple vertices in Figure 1), and "Others" (red vertices in Figure 1).

As Table 1 shows, the group with the highest number of respondents is the forestry and hunting sector (40.72%), followed by the agriculture sector (26.95%) and nature conservation sector (23.35%), which is in contradiction with the number of stakeholders actually included in the formulation process of PUN, in the case of the forestry and hunting and agriculture sector. Therefore, the forestry sector is overrepresented in our study when compared to the agricultural sector. The sectors with the lowest number of respondents are the fishery and water sector (4.79%) and "Others" (4.19%).

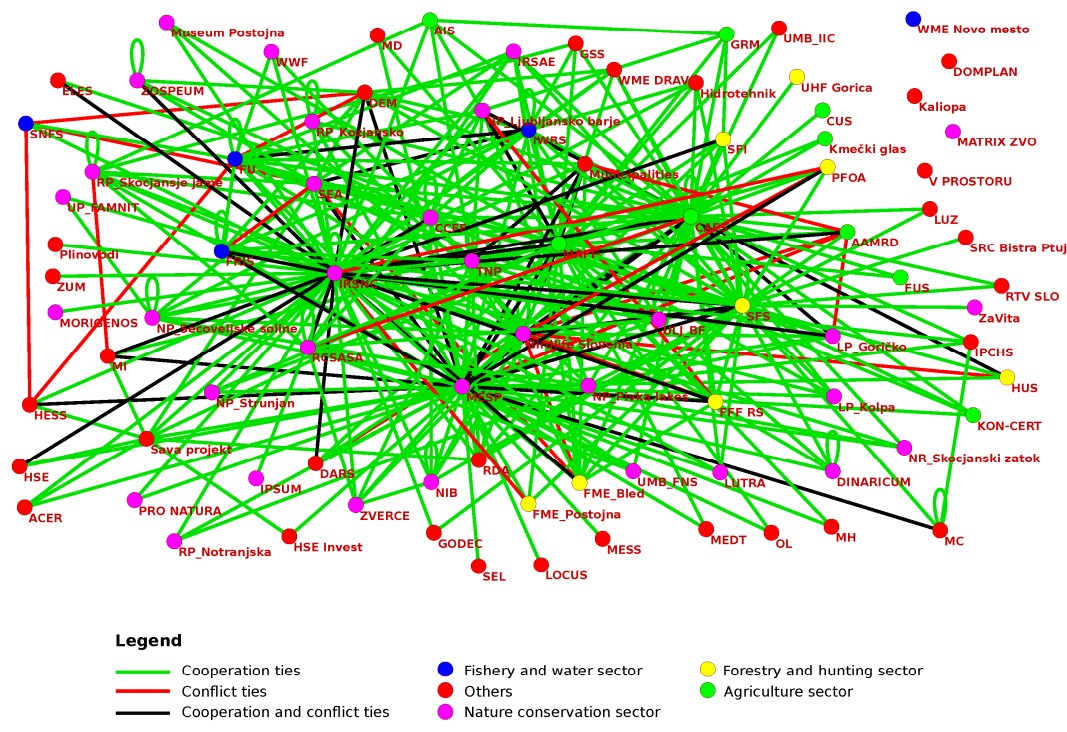

**Figure 1.** Cooperation and conflict network of the institutions in participatory process of formulation of Natura 2000 Management programme in Slovenia for the period 2015–2020 (print in colour).

*3.2. Cooperation and Conflict Network of Institutions – Locating Central Institutions*

Figure 1 shows the cooperation and conflicts networks as an undirected network between 88 institutions (some institutions are united as one institution, due to their small number of stakeholders, e.g., the municipalities, the Regional development agencies (RDA), or the fishery associations in the Fishery union), of which 5,68% had no ties with others. The network has 360 edges (undirected ties) in total, of which there are 299 edges that represent cooperation (coloured green in Figure 1), 21 edges that represent both cooperation and conflicts between institutions (coloured black in Figure 1), and 40 edges that represent conflicts (coloured red in Figure 1). Consequently, the cooperation network has 320 edges, of which there are 22 loops (which means that there is cooperation between co-workers from the same institution). The average degree for the cooperation network was 7.45, which means that, on average, each institution was connected to more than seven others. In addition, the conflict network consists of 61 edges and 0 loops, with an average degree 1.38, meaning that, on average, each institution was in conflict with some more than one other institution.

As reported in Table A1 (which summarized the top 10 DC values of institutions in the undirected network of cooperation), the institution with the highest DC is the IRSNC (DC = 57), followed by the MESP (DC = 55), CAFS (DC = 47), the Ministry of Agriculture, Forestry, and Food (MAFF) (DC = 44), SFS (DC = 38), and the NGO BirdLife Slovenia (DC = 18). Furthermore, a high number of institutions (35.23%) show rather low values of DC (DC lower than 2). Given these differences between the institutions that are related to the number of direct connections with others, it can be said that the cooperation network is a highly centralized network (see Figure 1), where one stakeholder (IRSNC) plays a key and central role.

BC was studied in order to obtain more detailed information on the cooperation structure of the network (see Table A2). The results showed that the central position in the cooperation network again belongs to the IRSNC (BC = 985.06), followed by the MESP (BC = 884.61), CAFS (BC = 585.87), MAFF

(BC = 451.97), and SFS (BC = 386.92). Institutions with higher BC values were potentially more inclined to play the role of intermediator.

A careful comparison of the DC and BC values showed that some central institutions (IRSNC, MESP CAFS, MAFF, and SFS) in the cooperation network had strong, immediate ties with various other institutions, and exercised a "bridging" function in the network.

Furthermore, the analysis of the DC, with its subdivision into IDC, was also calculated for the directed cooperation network (see Table A3). IRSNC was the institution with the highest concentration of power, which is mainly explained by the highest values of IDC, namely for IRSNC (IDC = 22), followed by MESP (IDC = 17), the NGO BirdLife Slovenia and the Slovenian Environment Agency (SEA) (for both institutions IDC = 14, respectively), and the Centre for Cartography of Fauna and Flora (IDC = 13).

The analysis of the conflict network between institutions (see Table A4) showed that the institution with the highest DC was IRSNC (DC = 19), followed by MESP (DC = 13), the NGO BirdLife Slovenia (DC = 9), CAFS (DC = 7), SEA, MAFF, and the Agency for agriculture markets and rural development (AAMRD) (for all three institutions DC = 5, respectively). The results are not surprising in the case of IRSNC and MESP, as they were the most active powerful in the cooperation network and, thus, in contact with many other stakeholders from different institutions and sectors. Each contact has the potential to bring new conflict. The NGO BirdLife Slovenia is one of the most powerful NGOs in this process, which has fully represented and argued its position in a few different sectoral workshops. Consequently, they came into conflict with other institutions with opposing views. The CAFS was a representative of landowners/farmers who have strong reservations about Natura 2000, because they were affected by the new forms of farming, management, and restrictions, and were excluded from the Natura 2000 sites designation phase. The SEA is responsible, among others, for the implementation of legislation and administrative procedures in the case of nature conservation conditions and permits for the construction of facilities and assessing the acceptability of these facilities as interventions in Natura 2000 sites. They have been in conflict with other institutions due to this responsibility. The MAFF is as the Ministry the charge of the agriculture, forestry, and fishery sector. In the process of formulation of PUN, MAFF participated in the intragovernmental consultation of the amended draft of PUN. They did not give their consensus to PUN until the draft PUN was harmonized with the forestry sector (SFS as a public forestry service). The AAMRD is an institution that is responsible for Rural Development Program implementation and it has been in conflict with other institutions due to their dissatisfaction with the financial incentives and compensatory measures.

*3.3. Frequency of Cooperation and Conflict among Sectors*

Table 2 shows the average cooperation network centrality values, while taking the different sectors into account. The nature conservation sector was a sector with, on average, the highest concentration of power (IDC = 5.88), followed by the fishery and water sector (IDC = 4.80), the forestry and hunting sector (IDC = 4.63), and the agriculture sector (IDC = 4.33) and the sector "others" (IDC = 2.55). The small differences in the values between the sectors show that all sectors had an impact and a significant role in the process of the formulation of PUN, but the impact was higher for sectors with IDC values above 4 than for the sector "others". The agriculture sector was most connected (DC = 12.78), followed by the nature conservation sector (DC = 9.70), forestry and hunting (DC =9.25), fishery and water sector (DC = 8.60), and the sector "others" (DC = 3.15). Regarding BC, the agriculture sector is the sector with the highest value of BC (BC = 115.96), followed by the nature conservation sector (BC = 62.05), the forestry and hunting sector (BC = 49.43), the fishery and water sector (BC = 10.80), and the sector "others" (BC = 1.28), respectively.

**Table 2.** Basic centrality values of the cooperation network by sectors.

| Sector | BC | DC | IDC |
|---|---|---|---|
| Forestry and hunting | 49.43 | 9.25 | 4.63 |
| Agriculture | 115.96 | 12.78 | 4.33 |
| Fishery and water | 10.80 | 8.60 | 4.88 |
| Nature conservation | 62.05 | 9.70 | 5.88 |
| "Others" | 1.28 | 3.15 | 2.55 |

The results on the frequency of cooperation between all institutions ($n = 88$) from different sectors in the participatory process of the formulation of PUN are very clear: institutions that belong to the same sector tend to show more mutual cooperation (Figure 2). The values presented in the matrix in Figure 2 represent the percentage of institutions from column clusters with which an average institution from row clusters repositions cooperation. The matrix is not symmetrical, as cooperation is measured as an asymmetric (directed) relation. The cooperation process in the case of the formulation of PUN was clearly fragmented across sectors. For example, institutions that represent the fishery and water sector hardly interact with other sectors and vice versa. Most of the cooperation of this sector was with institutions within their sector and far less with institutions within the nature conservation sector. Additionally, institutions from the forestry and hunting sector mostly interacted with institutions within their sector, while the agriculture and nature conservation sector more often interacted with institutions from other sectors.

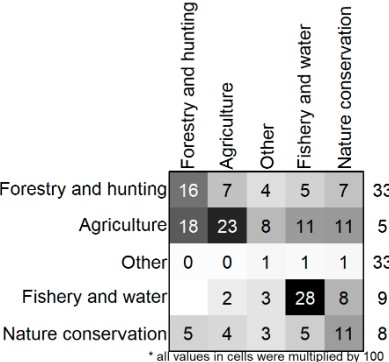

**Figure 2.** The matrix of cooperation among sectors in the participatory process of formulation of PUN.

The conflict between the institutions in the participatory process of the formulation of PUN appears between institutions from different sectors (Figure 3). The strongest conflicts occurred in the forestry and hunting sector and in the nature conservation sector, as well as between the agriculture and the nature conservation sector. Less frequent conflicts occurred between the agriculture sector and the forestry and hunting sector. The exceptions were institutions from the fishery and water sectors. This sector seems to have had many conflicts within its own sector, as well as conflicts with "other" sectors (i.e., Power plant Maribor—DEM and Water power plant Sava—HESS), which are related to the restrictions and limitations on the construction of dams for hydro-electric power, fish passes, and nature conservation sector.

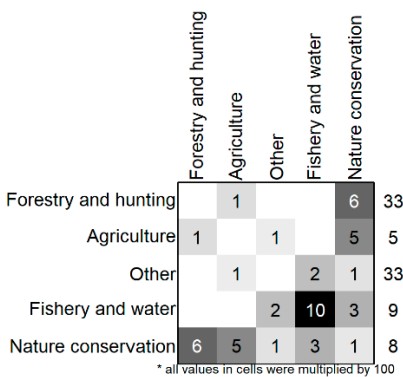

**Figure 3.** The matrix of conflicts among sectors in the participatory process of the formulation of PUN.

### 3.4. Partitioning the Institutions Based on Their Cooperation Network

The results partitioning the institutions based on their ties in the cooperation network are presented in the form of a partitioned matrix (Figure 4). The black square in the matrix indicates that the column institution reported cooperation with the row institution. The institutions that participated in the survey have black labels and they are positioned in the top and left part of the matrix. The institutions that did not participate in the survey have red labels and are only positioned in columns in the right part of the matrix. Since they have not responded to the survey, there is no row entry for them. The blue lines partition the units into clusters and ties into blocks. The thicker blue lines separate the participating and non-participating units.

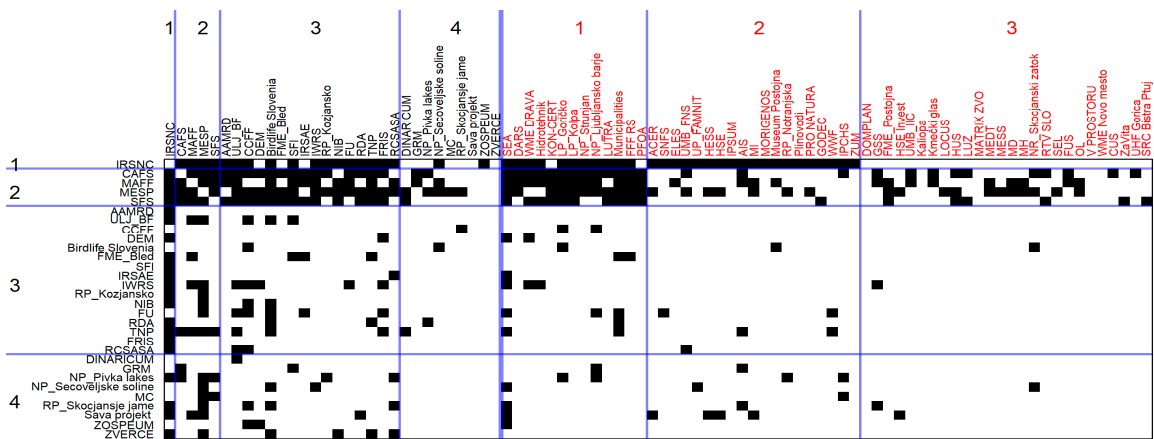

**Figure 4.** Results of homogeneity generalized blockmodeling of the cooperation network with removed isolates (print in colour).

Based on blockmodeling, four groups of institutions whose stakeholders participated in survey and three groups of institutions whose stakeholders did not participate in the survey were identified. These were identified and named according to their cooperation network. First, we will look at the clusters of participating institutions. Cluster 1 is well-connected with incoming and outgoing ties to most other clusters and, therefore, it was labelled "core institutions". Cluster 2 was named "semi-core institutions", while internally it is well-connected and has a lot of outgoing ties and has much less incoming ties from other clusters. Cluster 3 was named "semi-periphery institutions", as it is internally very sparsely connected and generally has fewer ties than previous clusters. Cluster 4 was named "periphery institutions" due to having the fewest ties, especially no ties within itself.

Cluster 1 only contains one institution, but it is the most important—IRSNC. This institution reported ties to: a) all institutions from cluster 2 of the participants; b) to most institutions from cluster 3

(except for the DEM, Forest enterprise Bled—FME Bled, and the Inspectorate of the Republic of Slovenia for Agriculture, Forestry, Food, and the Environment—IRSAE); c) to almost all institutions from cluster 1 of the non-participants (except for the Institute for inspection and certification in agriculture and forestry in Maribor—KON-CERT and the Private forest owners association—PFOA); and, d) to all institutions from cluster 2 of the non-participants. They also reported ties to some of the institutions from cluster 3 of the participants (i.e., IRSNC cooperated with NP Sečoveljske soline and ZOSPEUM), while they are unconnected with institutions from the last cluster of non-participants. On the other hand, most other institutions participating in the network reported cooperation with them.

Cluster 2 included four institutions that are fully connected. They are also well connected with institutions from cluster 1 of non-participants (i.e., MAFF from cluster 2 is connected to all institutions from cluster 1 of non-participants, except LUTRA, and CAFS is connected to all of them), with IRSNC from cluster 1 of participants and with cluster 3 from participants (i.e., SFS is connected with all of them, except with RP Kozjansko and FU), while most of them also report some ties with institutions from other clusters.

The 16 institutions from cluster 3 of participants have a relatively low number of incoming ties (some of them also have none), while most of them reported ties with IRSNC from clusters 1, with cluster 2 of the participants (i.e., CAFS, MAFF, MESP, and SFS), with cluster 3 of the participant (i.e., ULJ_BF, CCFF, NGO BirdLife Slovenia, FRIS, TNP, etc.) and cluster 1 of the non-participants (i.e., SEA, Hidrotehnik, WME Drava, LP Goričko, Lutra, etc.), as well as some sporadic ties to institutions from other clusters.

The nine institutions from cluster 4 of the participants have a relatively low number of incoming ties (some also none—i.e., ZVERCE), while most of them reported ties with institutions from cluster 2 of the participants (i.e., CAFS, MESP, MAFF, and SFS), and some of them with IRSNC from cluster 1 of the participants. They have some sporadic ties to institutions from other clusters and no ties to institutions from cluster 4 of the participants.

The incoming ties of cluster 1 of non-participants (non-participants could not report outgoing ties, although most of them probably have them) are very similar to the incoming ties of cluster 3 of participants. The core institution (IRSNC) reported ties to all institutions in this cluster (except KON-CERT and PFOA) and most other institutions from cluster 2 of participants reported ties to at least some institutions in this cluster. They have only sporadic incoming ties from other institutions. We assume that their outgoing ties would match those from participants of the semi-periphery, had they reported them.

A core institution named each unit from the second non-participants cluster, while they only have sporadic incoming ties from other clusters. In this sense, they are a true periphery. Institutions from the third non-participants cluster have some ties from "semi-core institutions", but other clusters very rarely mention them.

## 4. Discussions

Nature conservation policy and decision-making on nature conservation issues are characterized by complex interactions between different stakeholders from different institutions and sectors [13] in the EU. Traditionally, the nature conservation policy in Slovenia has been based on a centralized top-down approach whit decision–making power distributed among a few institutions [48]. With the implementation of the Natura 2000 network and the formulation of the management programme—PUN in Slovenia, a participatory approach has spread in nature conservation policies to respond to the need to involve stakeholders from different institutions that are essential for the fulfilment of EU nature conservation objectives.

A detailed stakeholder analysis of the participatory process of the formulation of PUN in Slovenia showed many individual stakeholders from different institutions and sectors who participated in the process facilitated the process. The highest number of stakeholders came from forestry and hunting, followed by agriculture and the nature conservation sector. This can be explained by the fact that

SFS and CAFS are institutions with a large number of stakeholders in the forestry and agricultural sector. Despite the fact that many stakeholders participated in the PUN, only one institution had a key role—IRSNC—and a few institutions had important roles (CAFS, MAFF, MESP, and SFS) in the decision-making process, while some of the institutions had a peripheral position, either due to their willingness not to participate in a participatory process or because of the lack of information and opportunities to join the network. Furthermore, the results of previous studies show Manolache et al. [56] and Blicharska et al. reported similar findings [37], where the participatory process was centralized around a small number of public authorities, with a low level of participation by NGOs and private stakeholders. Moreover, the same studies found a lower level of cooperation between stakeholders in the network periphery with other institutions, which indicates the clear top-down approach of the participatory process. In our case, landowners and farmers had a central representative (CAFS) with an important role in the process, while other representatives of them were on the periphery of the network (i.e., PFOA). Additionally, in our case, NGO involvement was not as low as reported in previous studies, but they had less power to influence the process than the public authorities.

From the results of the SNA analysis, it can be concluded that the cooperation network of the participatory process of the formulation of PUN is a highly centralized network. In the cooperation network, one institution (IRSNC) is dominant and in a central position; consequently, with the highest power in the decision-making process, while the ministries (MESP and MAFF), CAFS, and SFS were in a sub-dominant position, but still central position. The dominant and central position of the IRSNC in a cooperation network is not surprising, since the IRSNC, as a national expert institution for nature conservation, was ultimately able to organize workshops and coordinate nature conservation measures among stakeholders. Besides the IRSNC, MESP, MAFF, and CAFS played important roles in transferring information and, consequently, in the PUN influencing decision-making process. MESP was primarily responsible for the content and financial management of the entire project, for informing the partners and monitoring the progress of the activities and providing professional guidance to the partners. The MAFF, as a ministry in charge of agriculture, forestry, and fishery sector, participated in intragovernmental consultation of the amended draft of PUN, while CAFS participated in the network as representative of farmers and landowners, as a project partner. Influential institutions were, in addition to those already mentioned (IRSNC, MESP), the NGO BirdLife Slovenia and CCFF, both of which were involved as external experts on birds, flora, and fauna. All of these institutions were very powerful and, consequently, important, due to their roles in the process of formulation of PUN. The results were confirmed by a previous study undertaken by Laktić and Pezdevšek Malovrh [38], in which the stakeholders stated, during the face-to-face interviews, that the ultimate decision-making power in the PUN process lies with MESP and IRSNC. Anyhow, the results might be contradictory with the logic of bargaining theory, where it is argued that a central position of institution can, in fact, undermine an institution's (bargaining) power, e.g., if other (peripheral) institutions have influence, but only little interest in an outcome. Therefore, a central position does not guarantee the highest power [73].

Furthermore, the results of the blockmodeling confirmed the results of the SNA analysis, as the IRSNC was perceived as the most important institution in the participatory process of the formulation of PUN. According to the results of the blockmodeling, the IRSNC was cooperating with almost all institutions from different clusters—cluster 2, cluster 3, cluster 1 of non-participants and cluster 2 of non-participants. In addition to IRSNC, CAFS, MAFF, MESP, and SFS are also considered as important actors in mobilizing the network and bringing together other stakeholders from different institutions. However, these stakeholders have to exert a lot of energy to maintain a large number of ties and, therefore, these ties are often weak [47]. Moreover, central institutions are crucial for a participatory process, as their links are used to disseminate information and possibly mobilize other stakeholders from different institutions to act, although there is no guarantee that they can significantly influence those to whom they are tied. On the other hand, it can be said that these institutions have had control over information and communication, and have been able to restrict the communication/participation

of others. This situation has led to information asymmetry in the formulation of PUN and the exclusion of some institutions from the decision-making process. Therefore, the weakest point of the cooperation network was related to the imminent exclusion of peripheral institutions from the decision-making process. Therefore, these institutions had no central role and power to influence the decision-making process. Therefore, others might have voiced their interests and points of view. According to [63], this lack strengthens the institutional power of others and can lead to a decrease in inclusiveness, partial delegitimization, and a general weakening of the decision-making process. If a stakeholder is weakly or not at all connected to other stakeholders, this does not mean that his or her opinion is unimportant [55]. Therefore, it is suggested to involve these stakeholders more actively in the dialogue and decision-making process on nature conservation policy in the future.

The Natura 2000 network has important implication on many sectors competing for or using the same land. Therefore, Natura 2000 management plans must take the interests and views of a large number of stakeholders from different institutions and sectors into account. In addition, Natura 2000 management plans should be well integrated into other sectors and policies, as they are directly or indirectly affected by decisions taken within the Natura 2000. As a result conflicts of interest arise. The conflict network showed a similar position of the institutions in the participatory process of the formulation PUN as a cooperation network. IRSNC and MESP were the institutions with central positions in a conflict network, as they lead and organized the PUN process, due to their role in the nature conservation system in Slovenia. Thus, they were in contact with many other stakeholders from different institutions and sectors. Each contact has the potential to bring a new conflict. Among IRSNC and MESP, there is also an NGO, BirdLife Slovenia, and representatives of farmers and landowners (CAFS), who took a central position in the conflict network. The NGO BirdLife Slovenia was very active in the cooperation network, where it had many contacts with other institutions and, consequently, often came into conflicts with others due to different opinions on conservation measures related to birds. In addition, the NGO BirdLife Slovenia participated in various sectoral workshops, which led to conflicts with different sectors. CAFS, representing farmers and landowners, was in conflict mainly because of the amount of compensation for payments for restrictions in Rural Development Programme and the reduction of the number of allowed cuttings due to the delayed harvest date.

According to the structure of the cooperation and conflict networks, in the case of PUN, it can be said that it is a more favorable structure for a top-down decision-making process than a bottom-up one, which means that the transformation from a state government dominating the nature conservation approach to a more collaborative governance form might not be enduring. This is not surprising, as Slovenia has been traditionally dominated by a policy of command and control with dominated state institutions in the past, and this still affects the institutional capacity to develop an adequate participatory process for interaction between stakeholders from different levels, which consequently influences the legitimacy of the decision-making process. Many studies provided evidence that a top-down implementation approach prevailed in most EU countries, which causes conflicts with various stakeholders (mainly land users) who felt excluded [13,14]. This led to a "participatory" shift towards more socially inclusive and participatory bottom-up approaches, as reported in Weiss et al. [13].

The results regarding cross-sectoral cooperation between institutions in the participatory process of the formulation of PUN were predictable. The cooperation process in the formulation of PUN in Slovenia was clearly cross-sectoral, as institutions that belong to the same sector tend to have more mutual cooperation. The reason for the sectoral fragmentation in our research lies in the fact that workshops (round tables) and meetings for institutions within a sector were separately organized, which ensures the intra-sectoral exchange of knowledge and enables the sectoral prioritization of objectives. According to the results, the nature conservation sector was in a dominant position with the greatest influence on framing the policy process of the formulation of PUN, as Natura 2000 is a core pillar of nature conservation policy in the EU. As many stakeholders from different institutions and sectors were involved in the participatory process of the formulation of PUN, a lot of time and resources were needed to design the policy, often without achieving a satisfactory compromise between

the different interests, as there are advocacy coalitions networks organized around shared normative commitments and causal beliefs in each political subsystem [74]. These coalitions are resistant to change and conflicts between them can create polarized policy problems. According to the results of our study, such problems occur, for example, between forestry and nature conservation coalitions over the amount of deadwood and the restrictions of the construction of forest roads between agriculture and nature conservation over appropriate mowing periods or over Natura 2000 payments in the Rural development programme. Previous studies have often shown that nature conservation and forestry coalitions are in opposition to each with respect to the implementation of Natura 2000 [13]. For instance, Ferlin et al. [75] found that the cause of the conflict between nature conservation and the forestry sector in Slovenia can be traced to divergent conceptions of what is "appropriate" nature conservation in forestry—either segregation from areas that are actively managed (as seen by the MESP) or integrated activities within sustainable management, as seen by SFS and MAFF. Moreover, many studies describe conflicts between competing nature conservation and land use policy advocacy coalitions and provided evidence that land use policy coalitions have resisted expansion of the Natura 2000 network and increased enforcement in many EU countries [34–36,76–78]. Thus, the pro and contra coalitions appear to exist in most EU Member States [13]. In our study, the exception was the fishery and water sector, where there appeared to be many more intra-sectoral conflicts than conflicts with other sectors. In this case, intra-sectoral conflicts could merge due to the fact that, in the process of the formulation of PUN, institutions from the fishery sector and institutions from the water sector were merged into one sector. Another explanation could be found in the fact that different legal and policy frameworks specific for each sector regulate both sectors separately. It is therefore expected that each sector will have to ensure the implementation of its own objectives and measures and represent different interests and values that may cause conflict.

However, policy enforcement of PUN and practical management at local level might be limited due to the observed structure of the cooperation network, where the lack of power and involvement of stakeholders from the local level (landowners' representatives, except CAFS, land users, municipalities, rural development agencies, and other concerned social groups) is noticed. Moreover, some institutions of the network took a peripheral position. This finding is in line with the study done by Laktić and Pezdevšek Malovrh [38], which show that the main limit of the PUN was related to stakeholders' involvement, where some groups of stakeholders felt excluded from the process or were only involved only in the final stage. They argued that one reason for that could be found in the fact that only the interests of privileged stakeholders were included in the decision-making process, while other marginalized stakeholders were only informed regarding the results of the process in the final stage. For policy decision-makers, it is necessary to balance the involvement of all stakeholders' groups from different institutions and to combine participation with co-responsibility in order to improve the perceived influence of the institutions. Especially, as more frequent participation of local level stakeholders is needed in the initial phase and should be based on consensus decision-making and involvement is a good example of this, which would improve future PUN participation processes. The Austrian Forest Forum, where landowners have had the opportunity to participate in the dialog process via an internet platform or through written statements. In addition, landowners in the Austrian Forest Forum were kept informed through a Forest Dialog Newsletter [79].

When interpreting the results of this study, it should be noted that the approach has some limitations. The SNA approach in this study is based on whether or not the connections/ties between institutions exist. The strength of these connections was not considered, although it was recorded, because the strength of the tie was reported at individual (and not organizational) level and as most reported ties very similar strength (participants did not report weak ties). Therefore, tie strengths were not taken into account; it use would not add much information, while it would significantly increase the complexity of the presented results. The analysis only considers the cooperation and conflict network of the formulation of PUN of stakeholders from different institutions that participated in the survey. Based on the responses of the respondents (*n* = 167), the present study does not reveal

the opinions of the silent majority, or their future behaviour. Due to confidentiality concerns, the non-respondents were not followed up, so that the differences with the respondents were not estimated. However, the number of respondents in the survey was large enough to represent an important part of the network. The 167 stakeholders came from 34 different institutions, representing 38.64% of all (88) institutions involved in the process of the formulation of PUN. The representativeness of the sample was checked by examining the distribution of respondents to see whether they were randomly distributed across sectors. As shown in Table 1, responses of the forestry sector are over-represented when compared with the agriculture sector. However, the distribution across categories of stakeholders from different sectors can be considered to be comparable, as it is able to take into account the interests of all sectors (Table 1). In addition, the authors consider that the SNA is sufficient to identify relevant stakeholders that are located in the core of the network or that represent specific group categories in the network. After all, 167 stakeholders have revealed with whom they were involved in the process of the formulation of PUN. Therefore, their answers reveal not only their activities or only their ties, but also those with whom they cooperated. Since the importance of stakeholders in the network is primarily judged by how they are seen by others, we were able to identify the most important stakeholders, even if they (or their employees) did not participate in the survey. With regard to the methodology adopted and the survey questionnaires, SNA have the advantage of being simple and easy to apply. The limitations of the survey methods applied are typical for questionnaires, such as incomplete answers or the low reliability of answers to certain questions. Another example of the limitations in this study is the classification of stakeholders and their institutions into sectors. Some institutions, such as MESP, MAFF, and ULJ BF, cover more than one sector. As the survey was anonymous, the respondents were asked to indicate their institution. The institutions indicated were grouped into sectors according to the criteria of dominance of stakeholders from a particular sector. For example, most of the MAFF stakeholders work in the agriculture sector and less in the forestry sector. Therefore, in our study, MAFF is grouped with the agriculture sector. The limitations of SNA and blockmodeling are also that the number of respondents influences the results. In this case, SFS and CAFS are institutions with many stakeholders (employees) involved in the process of the formulation of PUN and also many of them have responded to our survey. Consequently, their importance in the process might be somewhat exaggerated. Therefore, the analysis should be extended in the future studies to include a qualitative assessment of the network. A combination of the qualitative and quantitative approach would provide better insight into stakeholder participation in the case of the formulation of PUN.

## 5. Conclusions

In recent decades, the number of studies analysing participatory processes in natural resource management have rapidly increased [8,22,26,35,36,41,46,55,57,74,80,81]. Nevertheless, the analysis of the relationship between stakeholders and decisions taken during the formulation process of Natura 2000 management plans remains within the scope of the study. The present study attempts to contribute to the scientific debate in this area, focusing on the stakeholders' analysis and relationships (cooperation and conflicts) between the stakeholders during the participatory process of the formulation of Natura 2000 management programme in Slovenia. The results identify institutions and sectors that have a central position in a cooperation and conflict network and, thus, have the highest power to influence a decision-making process. In addition, institutions that acted in isolation and passively participated in the formulation of the Natura 2000 management programme or were separated from the network. Understanding the participation planning failures and integrating the perspectives of the periphery or marginalised group of stakeholders from different institutions into the future participatory processes of the formulation of Natura 2000 management programme could lead to better management plan enforcement at the local level and defuse potential conflicts. Therefore, it is important for policy decision makers to empower different groups of stakeholders from different institutions and sectors in further participatory processes and balance the interventions of different groups of stakeholders during the negotiation process. Furthermore, the study showed the usefulness of the methodology in

identifying relevant stakeholders and institutions that are involved in the decision-making process of the formulation of PUN.

**Author Contributions:** The results and article are part of the emerging of T.L. and his supervisor Š.P.M., and co-supervisor T.K., with the title: »The Characteristics of the Social Networks and Participation of Stakeholders in the Management of Natura 2000 Sites«.K., T.L: conceptualization, data curation, writing original draft preparation; A.Ž.: software, data curation and formal analysis, editing; T.K.: conceptualization, supervision and editing; Š.P.M.: conceptualization, writing original draft preparation, writing- reviewing and editing, supervision. All authors have read and agreed to the published version of the manuscript.

**Funding:** This research was funded by the Pahernik foundation. The authors wish to thank the foundation for supporting the publishing of the results.

**Acknowledgments:** Authors wish to thank all the respondents who took part in this research and made it possible by sharing their experiences.

**Conflicts of Interest:** The authors declare no conflict of interest. The funders had no role in the design of the study; in the collection, analyses, or interpretation of data; in the writing of the manuscript, or in the decision to publish the results.

## Appendix A

**Table A1.** Top 10 in the degree of centrality values (DC) for institutions in the undirected network of cooperation.

| Nº | Institution (Acronym) | Sector | DC |
|----|----------------------|--------|-----|
| 1 | IRSNC | Nature conservation | 57 |
| 2 | MESP | Nature conservation | 55 |
| 3 | CAFS | Agriculture | 47 |
| 4 | MAFF | Agriculture | 44 |
| 5 | SFS | Forestry and hunting | 38 |
| 6 | NGO BirdLife Slovenia | Nature conservation | 18 |
| 7 | TNP | Nature conservation | 17 |
| 8 | FU, IWRS | Fishery and water | 16 |
| 9 | CCFF | Nature conservation | 15 |
| 10 | ULJ_BF, SEA | Nature conservation respectively | 14 |

**Table A2.** Top 10 in the betweenness centrality values (BC) for institutions in the undirected network of cooperation.

| Nº | Institution (Acronym) | Sector | BC |
|----|----------------------|--------|-----|
| 1 | IRSNC | Nature conservation | 985.06 |
| 2 | MESP | Nature conservation | 884.61 |
| 3 | CAFS | Agriculture | 585.87 |
| 4 | MAFF | Agriculture | 451.97 |
| 5 | SFS | Forestry and hunting | 386.92 |
| 6 | TNP | Nature conservation | 34.55 |
| 7 | FU | Fishery and water | 33.79 |
| 8 | NGO BirdLife Slovenia | Nature conservation | 32.10 |
| 9 | SEA | Nature conservation | 22.84 |
| 10 | Sava projekt | Other | 19.82 |

**Table A3.** Top 10 indegree centrality values (IDC) for institutions in directed network of cooperation.

| Nº | Institution (Acronym) | Sector | IDC |
|---|---|---|---|
| 1 | IRSNC | Nature conservation | 22 |
| 2 | MESP | Nature conservation | 17 |
| 3 | NGO BirdLife Slovenia, SEA | Nature conservation respectively | 14 |
| 4 | CCFF | Nature conservation | 13 |
| 5 | ULJ_BF | Nature conservation | 11 |
| 6 | MAFF | Agriculture | 10 |
| 7 | FRIS, Municipalities, TNP | Fishery and water, Nature conservation, Other | 9 |
| 8 | SFS, RCSASA, NP Ljubljansko barje, NP Goričko CAFS, SFI | Forestry and hunting, Nature conservation, Nature conservation, Nature conservation, Agriculture, Forestry and hunting | 8 |
| 9 | IWRS | Fishery and water | 7 |
| 10 | FFFRS, RDA, FU, NIB, AIS, WME DRAVA | Forestry and hunting, other, Fishery and water, Nature conservation, Agriculture and, other | 6 |

**Table A4.** Top 10 in the degree of centrality values (DC) for institutions in undirected network of conflicts.

| Nº | Institution (Acronym) | Sector | DC |
|---|---|---|---|
| 1 | IRSNC | Nature conservation | 19 |
| 2 | MESP | Nature conservation | 13 |
| 3 | NGO BirdLife Slovenia | Nature conservation | 9 |
| 4 | CAFS | Agriculture | 7 |
| 5 | SEA, MAFF, AAMRD | Nature conservation, Agriculture, Agriculture | 5 |
| 6 | Municipalities, FFF RS, DEM | Other, Forestry and hunting, Other | 4 |
| 7 | HESS, SNFS, FME Bled, NP Goričko, NP Ljubljansko barje, MI, FU, SFS, PFOA, FRIS | Other, Fishery and water, Forestry and hunting, Nature conservation, Nature conservation, other, Fishery and water, Forestry and hunting, Forestry and hunting, Fishery and water | 3 |
| 8 | DARS, IWRS, HUS, RCSASA | Other, Fishery and water, Forestry and hunting, Fishery and water, Nature conservation | 2 |
| 9 | CCFF, ELES, FME Postojna, SFI, HSE, MC, RP Škocjanske jame, TNP, ZOSPEUM | Nature conservation, Other, Forestry and hunting, Forestry and hunting, Fishery and water, Other, Nature conservation, Nature conservation, Nature conservation | 1 |
| 10 | The Rest | | 0 |

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
