# Peer review of "Stakeholders’ Social Network in the Participatory Process of Formulation of Natura 2000 Management Programme in Slovenia"

_forests, doi:10.3390/f11030332_

Round 1
Reviewer 1 Report
The specific research focusing on the stakeholders analysis and cooperations and conflicts among stakeholders during their participation process of the formulation of Natura 2000 management programme. Although the conclusions extracted from the paper are interested but limited only for Slovenia. The particular work can be done, also to other European countries and the conclusions can be compared and connected for future cooperations between the sectors and institutions of the countries. The analysis could be extended with a qualitative evaluation of the total EuropEan network in future studies.
The only common for improvement is that the figure 4 is not giving clear results to the readers in my opinion it would be better to removed from the paper.
Author Response
Dear R1,
please find attached our answers.
Kindest regards,
Špela
Reviewer 2 Report
Reviewer Recommendation and Comments for Manuscript forests-743919: Stakeholders’ ‘Social Network in the Participatory Process of Formulation of Natura 2000 Management Programme in Slovenia
The study analyses communicative interrelations between agents involved in the formulation of Slovakia’s Natura2000 management programme, applying social network analysis (and related approaches). The analysis is sound and presented comprehensively; the English is fairly good (disregarding some minor coarseness which is, however, well acceptable, e.g. the frequent lack of definite articles). In my mind the article should be published after some minor revisions.
Some issues which might be worth considering
· As “participation” is no standardised phenomenon, it might be sensible to mention in the introduction that different degrees of participation can be distinguished (I’m sorry that I only remember some older literature at the moment; see e.g. Scharpf (1997) or Elsasser (2002) [quotation from the latter: “Several degrees of participation are possible, ranging from ‘passive’ participation (with the modest ambition that stakeholders be informed about decisions made by others, and the decision process thus become transparent) to ‘interactive’ participation which requires joint decisions, and perhaps shared liabilities”])
· Section 2.2 is a bit confusing. As I understood it, you had 815 participants in the PUN development in total; of these, you have not been able to contact X1 {unknown number} due to missing addresses, hence you contacted only X2; of those contacted, X3 refused taking part in the survey, (and X4 refused partly?), so that you ended up with 266 surveys available for analysis. As 99 of these did not answer completely, SNA can be conducted with 167 answers only.
· L 172-174: Unclear; you seem to mean that for a confidence level of 95 % you would have needed 261 answers. However, this depends on the target variable as well as on its variance. Since you seem to have several target variables, I’d suggest deleting this sentence (it seems over-generalised).
· Tab.1, suggestion: merge with tab.2 (i.e. add 2 columns for the available answers here)
· Section 2.3: specifically the first paragraph (l. 192-208) uses quite a lot of jargon which might not be fully understandable for readers not familiar with network analysis (undirected network, undirected representation, connected vertices (institutions), blockmodeling, single vertex (institution), degree and betweenness, degree centrality (DC), indegree centrality (IDC))
· Still section 2.3: It is not quite clear which data exactly you have used for this analysis (since you have described the questionnaire only in a general way), and hence your study would not be reproducible by another researcher. (This is my “most severe” criticism).
· Results, section 3.1 (in comparison to 2.2): it seems that answers of the forestry sector are overrepresented while the agri sector is underrepresented (as compared to the study population reported in Tab.1); this should be mentioned shortly here. (Additionally, Tab.2 should be merged with tab.1, as mentioned above)
· Section 3.4: (Just as a comment for the editor: Listing all these national institutions might seem too specific for an international audience – however, it is necessary in order to understand the analysis at which figure 4 is based. Therefore, if other reviewers suggest omitting this level of detail: I suggest keeping it!)
· Line 461-462: “not” taken into account is too strict as an interpretation, as their interests might have been voiced by others
· A point possibly worth discussing shortly (in the discussion): the approach is based at measuring the number of connections/ties between institutions, but not necessarily the strength of these connections.
· Line 594, “The results identify institutions and sectors that had a central position in a cooperation and conflict network, consequently with the highest power to influence a decision-making process”: a game-theoretic analysis might come to the opposite conclusion: a central position can in fact undermine an institution’s (bargaining) power, e.g. if other (peripheral) institutions have influence, but only little interest in an outcome. At least, a central position does not guarantee highest power. (On this, see the literature quoted above, or e.g. Muthoo (1999) – again quite old, but still up-to-date).
Specific comments
· Line 74, “scientific” criteria vs. social criteria: do you mean “physical”, or “directly measurable” rather than “scientific”?
· L 80, “opposed” rather than “contradictory”?
· L 87: As identifying “all” stakeholders might be impossible, I’d rather formulate “a technique that intends to identify all groups of stakeholders…”
· L 133, “subordinate agencies” rather than “associated beneficiaries (partners)”?
· L 145 Who has prepared this communication plan (and the PUN draft mentioned in line 148)?
· L 149 this list seems a bit unsorted; perhaps subsume as “additional experts” (and possibly add in brackets for which subject areas they have been experts)
· L 151-154 this seems too unspecified here; therefore you may want to add a cross reference (to section 2.2)
· L 159 the “new” PUN: confusing, as you just have mentioned one single PUN before
· L 160/161 what is the difference between “mere information-communication” and “consultation”?
· L 171: “belong to” rather than “represent”
· L 181, six sections: in the following you mention only 4
· L 205, What exactly is a “concept of prestige”?
· L 235, 167 stakeholders: the reader expects 299 here, from what you said in section 2.2.
· Figure 1: please add explanation of colours to legend (only explained in the text, lines 251-254)
· Figure 2: please explain why numbers above/below the diagonal differ (or, in other words, what rows vs. columns mean)
· Figure 4: Cluster numbers should be added. – Beyond that, the texts are barely readable (too small), and the bottom part is uninformative and wastes much space. Therefore, I suggest summarising the whole lower part (shaded in red) to one or three single rows which just list the names of the respective (non-responding) institutions, rather than reserving a separate row for each institution
· L 347, “non-existent”: You don’t know whether ties do not exist, as the respective institutions have not answered the survey
· L 351, Four groups: figure 4 shows 7 groups
· L 352: sentence incomplete; I guess you mean “Cluster 1 is well-connected to most other clusters, and at the same time it is clearly distinct from all other clusters; therefore, it was labelled “core institution”.”
· L 356: fewest (rather than least)
· L 385: “although most of them probably have them”, I guess
· L 437 “in charge of” (rather than “in head of”)?
· L 486: “and society” should be deleted (confuses the sentence)
· L 490: “might not be enduring” rather than “was not enduring”?
· L 491 dominating rather than dominated
· L 556-557: rather, “to present an important section of the network”.
· L 562-564: not really, as forestry is overrepresented and agriculture is underrepresented (see comment above)
Minor comments (language)
· Generally, check for missing or unnecessary articles (“the” / “a”)
· Lines 51-54 check singular/plural (sites, plans)
· Line 57: stakeholder involvement
· Lines 68-70: Sentence too long and difficult to understand; the part in parentheses might be deleted
· L 76: …, participatory approaches have been ignored in many…
· L 95: to enhance stakeholder analysis, social network analysis (SNA) has been frequently used...
· L 99, and it can also help…
· L 112: using stakeholder analysis combined with SNA in order to a) identify main the institutions’ cooperation and conflict network…
· L 115 using a blockmodeling approach
· L 126 the identification of measures for achieving…
· L 273 A careful comparison of DC and BC values revealed that…
· L 293-296 sentence is not well understandable. Do you mean “administration” rather than “conducting administrative proceedings”?
· L 497 delete “EFI”
· L 502, word order “for the”
· L 567 “reveal”
· L 577 check sentence
· L 578 “accordingly” rather than “additionally”
Literature quoted
- Elsasser, P., 2002. Rules for participation and negotiation and their possible influence on the content of a national forest program. Forest Policy and Economics 4, 291-300.
- Muthoo, A., 1999. Bargaining Theory with Applications. University Press, Cambridge.
- Scharpf, F.W., 1997. Games Real Actors Play. Actor-centered institutionalism in policy research. Westview Press, Boulder.

Author Response
Dear R2,
please find attached our answers.
Kindest regards,
Špela
